# Cost Analysis for Patients with Ventilator-Associated Pneumonia in the Neonatal Intensive Care Unit

**DOI:** 10.3390/healthcare10060980

**Published:** 2022-05-25

**Authors:** Ralitsa Raycheva, Vanya Rangelova, Ani Kevorkyan

**Affiliations:** 1Department of Social Medicine and Public Health, Faculty of Public Health, Medical University of Plovdiv, 4002 Plovdiv, Bulgaria; dirdriem@gmail.com; 2Department of Epidemiology and Disaster Medicine, Faculty of Public Health, Medical University of Plovdiv, 4002 Plovdiv, Bulgaria; ani_kevorkian@mail.bg

**Keywords:** nosocomial infections, ventilator-associated pneumonia, hospital costs, attributive costs, economic burden, antibiotics

## Abstract

The concept of improving the quality and safety of healthcare is well known. However, a follow-up question is often asked about whether these improvements are cost-effective. The prevalence of nosocomial infections (NIs) in the neonatal intensive care unit (NICU) is approximately 30% in developing countries. Ventilator-associated pneumonia (VAP) is the second most common NI in the NICU. Reducing the incidence of NIs can offer patients better and safer treatment and at the same time can provide cost savings for hospitals and payers. The aim of the study is to assess the direct costs of VAP in the NICU. This is a prospective study, conducted between January 2017 and June 2018 in the NICU of University Hospital “St. George” Plovdiv, Bulgaria. During this period, 107 neonates were ventilated for more than 48 h and included in the study. The costs for the hospital stay are based on the records from the Accounting Database of the setting. The differences directly attributable to VAP are presented both as an absolute value and percentage, based on the difference between the values of the analyzed variables. There are no statistically significant differences between patients with and without VAP in terms of age, sex, APGAR score, time of admission after birth and survival. We confirmed differences between the median birth weight (U = 924, *p* = 0.045) and average gestational age (*t* = 2.14, *p* = 0.035) of the patients in the two study groups. The median length of stay (patient-days) for patients with VAP is 32 days, compared to 18 days for non-VAP patients (U = 1752, *p* < 0.001). The attributive hospital stay due to VAP is 14 days. The median hospital costs for patients with VAP are estimated at €3675.77, compared to the lower expenses of €2327.78 for non-VAP patients (U = 1791.5, *p* < 0.001). The median cost for antibiotic therapy for patients with VAP is €432.79, compared to €351.61 for patients without VAP (U = 1556, *p* = 0.024). Our analysis confirms the results of other studies that the increased length of hospital stays due to VAP results in an increase in hospital costs. VAP is particularly associated with prematurity, low birth weight and prolonged mechanical ventilation.

## 1. Introduction

The concept of improving the quality and safety of healthcare is well known. However, a follow-up question is often asked about whether these improvements are cost-effective. Due to the lack of reliable data to inform about quality and safety in healthcare, there are some hesitations to increasing investments, until the financial benefits are more clearly defined [1].

Healthcare is a dynamic industry where the assets (staff, technology, equipment) needed for success are becoming increasingly scarce and expensive. Despite rising care costs, pressure from payers to resist these increases continues to grow. Improving quality by reducing medical errors, length of stay and costs is an important alternative to scaling up and hiring more staff—key factors contributing to rising costs. Reducing the incidence of nosocomial infections can offer patients better and safer treatment, and at the same time can provide cost savings for hospitals and payers.

The incidence of nosocomial infections (NIs) in the neonatal intensive care unit (NICU) is approximately 30% and accounts for up to 40% of reported neonatal deaths in developing countries [2]. Neonatal hospital infections, in addition to being the cause of a significant number of perinatal, neonatal and postnatal deaths, are also associated with increased healthcare costs. This is because the hospitalization of infected neonates is up to threefold longer than that of non-infected children [3]. 

Ventilator-associated pneumonia (VAP) is hospital-acquired pneumonia that develops in patients who have been intubated and have received mechanical ventilation for at least 48 h [4]. It is the second most common nosocomial infection and has a major impact on neonatal morbidity, survival, hospital costs and length of stay in the NICU [2,5,6]. The incidence of VAP in the NICU is difficult to pinpoint, as it is difficult to distinguish between new or progressive radiographic infiltrates due to neonatal pneumonia or due to the exacerbation of bronchopulmonary dysplasia and frequent episodes of atelectasis [7]. VAP occurs at higher levels among extremely low birth weight infants and is a major risk factor for complications and death (RR: 3.4; 95% CI: 1.20 to 12.32) [8]. VAP increases the length of stay in ICUs and in the hospital, and this results in increased costs of hospitalization. A few studies have been conducted in pediatric intensive care units (PICU) that might help determine the extent of the problem. A study from Nicaragua [9] estimated the average cost of hospitalization for a patient in the PICU with VAP at $9686, and $3779 for non-VAP patients. Romero et al. [10] calculated $6174.89 for the treatment of one episode of VAP. Studies from Iran [11] and the USA [12] have identified the prolonged hospital stay as the main driver of the attributable costs of up to $1040 and $51,157, respectively. Locally, in our scientific literature, there is a monograph published in which the authors did a landscape review on the global financial burden due to the treatment costs of nosocomial infections in NICU, but there are no studies that specifically estimate the costs for the treatment of VAP in the Bulgarian NICUs [13]. The aim of the current study is to assess the direct costs of VAP in the neonatal intensive care unit. 

## 2. Materials and Methods

### 2.1. Study Period and Settings

This was a prospective study, conducted in the period between January 2017 and June 2018 in the NICU of University Hospital “St. George” Plovdiv, Bulgaria. The study was conducted in one hospital setting which is located in the second largest Bulgarian city, and the NICU of the hospital is a level 3 NICU with respect to the care provided for the patients. This neonatology unit is the only available option for the South-Central region population, which represents approximately 20% of the overall Bulgarian population [14]. The number of deliveries for the period of the study was 3306 (1700 in 2017, and 1606 in 2018). Additionally, 52 infants (23 in 2017, and 29 in 2018) were admitted for medical treatment from other hospitals. Overall, 352 neonates in 2017, and 343 in 2018 were admitted for intensive care treatment in the NICU. 

### 2.2. Definition and Identification of VAP

VAP was defined as such by the criteria of the German system for surveillance of nosocomial infections NEO-KISS [15] and the Center for Disease Control and Prevention (CDC) [16]. Additionally, the criteria for VAP from the Bulgarian Medical Standard for Prevention and Control of Nosocomial Infections have been used [17]. VAP was defined as a clinically unstable respiratory condition with at least 2 or more clinical, and laboratory signs and symptoms and chest X-ray findings showing new or progressive infiltration and isolation of a pathogen from the endotracheal aspirate. The clinical signs included elevated temperature >37.8 °C, hypothermia, frequent apnea/bradypnea/tachypnea, bradycardia <80 b/m and change in tracheal secretions—color, quantity. Laboratory findings included—CRP >10 mg/L, abnormal white blood cells count (Leu > 30,000/mcg or Leu < 5000/mcg) and thrombocytopenia (Thr < 150,000/uL).

### 2.3. Patient Characteristics

During the study period, 507 neonates were followed up prospectively. Of them, 107 neonates were ventilated for more than 48 h and were included in the study. Data on the demographic characteristics of the patients, underlying diseases, clinical symptoms, X-ray examinations, the incidence of VAP, etiological agents and antimicrobial susceptibility rates were recorded. Endotracheal intubation was performed by observing the standard precautions (sterile gowns, masks, laryngoscope blades and tubes) to ensure maintaining the sterility of equipment until use. Endotracheal suctioning was performed every 8 h and in case of a need for microbiological material for the examination. Closed systems for endotracheal suctioning were used in the NICU. In the NICU, there is a standard protocol for the empirical antibiotic treatment that was followed by all neonatologists. The administration and duration of additional antibiotic treatment depended on the individual needs of the infant’s clinical condition (results from microbiological testing-antibiogram).

### 2.4. Hospital Costs

The costs of the hospital stay were based on the records from the Accounting Database of the University Hospital “St. George”. This accounting system includes components for direct and indirect costs, which are distributed under the standards of the Bulgarian Accountancy Act of 2015. The costs for patient-days are divided into the following categories: food (inpatient and staff); medicines (medicines, medical supplies, blood and blood products, disinfectants, hygienic materials, other medical expenses); fuels and energy (water, electricity, heat, stationery, other materials); current repairs, other external services (laboratory services); amortization, salary expenses (salaries and other remunerations and payments); insurance costs (insurance for state social premiums and health insurance premiums, premiums for senior medical staff); expenses for taxes, fees and other similar payments; other expenses. Neonatal costs per day were calculated individually for each patient according to the date of admission/discharge from the ward, duration of ventilation, date of diagnosed VAP infection and antibiotics used. The individual costs were summed in each category (VAP/non-VAP patients) to calculate the total costs on average for each month for the overall period of the study for the NICU. Our study site is a third level NICU, which provides complex care for the smallest and most premature infants until reaching a stable condition. Some of our prospectively studied (VAP/non-VAP) infants were discharged home, whereas those born prematurely were transferred to another NICU in our city for additional care until reaching the weight threshold for hospital discharge. There were 22 infants who were reported dead. 

### 2.5. Directly Attributable to VAP Differences

The difference directly attributable to VAP is presented both as an absolute value and calculated as a percentage, based on the differences between the values of the analyzed variables—average hospital stay (average patient days), average treatment duration (antibiotics), average hospital costs, average hospital costs per day, average costs for antibiotics and average costs for antibiotics per day for patients with and without VAP. The initial estimations were completed in 2018 Bulgarian leva (BGN). In order to facilitate the comparisons with other studies, Bulgarian leva were converted to European currency (€) at a fixed exchange rate of 1.95583 leva to the euro (fixed rate maintained under the International Monetary Fund-led currency board arrangement, since 1999).

### 2.6. Statistical Methods

Quantitative variables were presented as the mean ± standard deviation (mean ± SD) or median (25th percentile; 75th percentile) based on the sample distribution. The variables were compared for differences using independent samples, a *t*-test or Mann–Whitney test, based on the normality of the distribution. The Shapiro–Wilk test was applied to inform about the distribution of the patients sampled. Qualitative variables were presented as numbers/totals and percentages (*n*, %), and a z-test was applied to compare two proportions. The *p*-values < 0.05 were considered statistically significant for all statistical tests. A statistical analysis of the data was performed using SPSS v.26 for Windows (IBM Corp. Released 2019. IBM Corp., Armonk, NY, USA). For all tests, a *p*-value < 0.05 indicated the statistical significance. 

## 3. Results

### 3.1. Demographic Characteristics of Patients

VAP was diagnosed in 33 (30.8%) out of 107 patients included in the study. In two of the infants with VAP during the hospital stay, sepsis was diagnosed as a secondary NI, and in another one there was conjunctivitis as a second HAI. We lack the information about the primary diagnosis of all the ventilated patients. The distribution of VAP patients by primary diagnosis was as follows: RDS (respiratory distress syndrome) in the neonatal period—18 neonates; congenital pneumonia—5 neonates; congenital heart disease—3 neonates; extreme prematurity—4 neonates; and birth asphyxia—3 neonates. The other 74 non-VAP neonates were used as a control group. Males were 56.1% of all studied neonates, with a median age at the admission of 1 day (25th percentile—1 day; 75th percentile—1 day).

Table 1 presents a comparison between patients by groups, with and without VAP, respectively. There were no statistically significant differences between VAP and non-VAP patients in terms of age, sex, APGAR score at the 1st minute and 5th minute, time of admission after birth and survival. We confirmed statistically significant differences between the median birth weight (U = 924, *p* = 0.045) and the average gestational age (*t* = 2.14, *p* = 0.035) of the patients in the two study groups. There was a significant proportion of children with and without VAP born prematurely (born before 37 gestation weeks), respectively—81.8% (*n* = 27) and 70.3% (*n* = 52) without statistical significance between the two groups (z = 1.3, *p* = 0.211). In the group of children with VAP, we observed a relatively high percentage of children born weighing <999 g (*n* = 9, 33.3%), and before 28 gestational weeks (*n* = 13, 48.1%).

### 3.2. Etiology of VAP

Microorganisms invading the respiratory tract may cause VAP. The prevailing causative agents of VAP in our study were from the Gram-negative microflora with leading microorganisms *Klebsiella pneumoniae ESBLs+*—27.3% (*n* = 18) and *Acinetobacter baumannii*—13.6% (*n* = 9) (Table 2). In 45.5% of the patients with VAP, polymicrobial flora was isolated.

Additionally, 66 blood cultures have been taken in the 33 neonates with VAP, and 2 of the patients have been diagnosed with sepsis. In the first infant, there were two positive blood cultures in which *Coagulase (-) Staphylococcus* was isolated. In the second infant, one positive blood culture in which *Enterococcus faecium* was isolated. 

### 3.3. Length of Stay (LOS)

Most of the patients were admitted to the NICU in the first 24 h after birth—96.3% (*n* = 103). The median length of stay (LOS)/patients days for patients with VAP was 32 days (25th percentile—19 days; 75th percentile—46 days) compared to 18 days (25th percentile—11 days; 75th percentile—27 days) for patients without VAP (U = 1752, *p* < 0.001). The attributive hospital stay due to VAP was 14 days. For the group of patients with VAP, the median hospital stay and duration of mechanical ventilation before VAP diagnosis was 8 days (25th percentile—6.5 days; 75th percentile—10.5 days). There was a statistically significant difference between the median duration in days of mechanical ventilation between the two groups: with VAP, 12 days versus 4 days in the group of patients without VAP (U = 2068.5., *p* < 0.001). Lethality rates in both groups were close (z = 0.4, *p* = 0.688) (Table 1).

### 3.4. Costs

The median of hospital costs for patients with VAP was estimated at €3675.77 (25th percentile—€2498.87, 75th percentile—€5146.35), compared to the statistically significant lower expenses €2327.78 (25th percentile—€1434.10, 75th percentile—€3226.83) for patients without VAP (U = 1791.5, *p* < 0.001). The median cost of antibiotic therapy for patients with VAP was €432.79 (25th percentile—€282.48, 75th percentile—€994.23), compared to €351.61 (25th percentile—€212.42, 75th percentile—€587.75) for patients without VAP (U = 1556, *p* = 0.024).

Table 3 summarizes the costs directly attributed to VAP. Initially, all of the total costs were higher in the group of non-VAP patients, where we had approximately 55% more neonates. In the next step, we calculated the average costs and estimated the directly attributed VAP difference for each. Obviously, VAP adds significant expenditures in the length of stay (patient-days), duration of antibiotic treatment and less burden on hospital costs and costs for antibiotics. 

## 4. Discussion

The present study examines the costs associated with patients in the NICU diagnosed with VAP based on clinical diagnosis, microbiological results and X-ray examination for the first time in Bulgaria, thus avoiding the limitations of diagnosis only by clinical criteria and facilitating the identification of directly relevant costs associated with the diagnosis.

The cost estimate follows the approach of numerous economic studies in the field that focus on the main determinants of cost: length of hospital stay (patient-days), total hospital costs and analysis of the cost of antibiotic therapy. In addition, our approach includes calculating the directly attributed costs in absolute value and as a percentage, resulting from VAP. 

The analysis demonstrated a statistically significant difference in both hospital costs and length of stay (patient-days), as well as in the costs of antibiotic therapy for patients with and without VAP. This is important because VAP is one of the most common nosocomial infections in patients in pediatric and neonatal ICU [18]. The outlined increase in the duration of hospital stays between VAP and non-VAP patients might be explained by the fact that the patients with VAP had a statistically significant difference in birth weight and average gestational age. Low birth weight and prematurity have already been confirmed as independent risk factors for VAP in a meta-analysis of observational studies [19]. Low birth weight and prematurity imply a longer hospital stay, whereas VAP as a complication additionally requires prolongation of the stay until the infection is treated. 

VAP is the most common indication for the initiation of empirical antibiotic therapy in the pediatric intensive care unit (PICU), accounting for nearly half of all antibiotic days [20]. The leading pathogens isolated in patients with VAP in our study were from the Gram-negative microflora, which is in accordance with previous studies [4,7]. Additionally, almost half of the VAP patients had polymicrobial flora, which can further explain the longer duration of the antibiotic treatment and the higher number of antibiotics used in this group. The balance between adequate treatment and avoiding overtreatment with antibiotics is a challenging task. Studies determining the optimal duration of antibiotic use are sparse [21].

The absolute differences between the total number of patient-days and antibiotic-days, as well as between the total hospital costs and the total costs for antibiotics, show higher values of each of the variables for patients without VAP, but without considering that patients with VAP are 55.4% fewer compared to them. For this reason, the difference directly attributed to VAP between the two groups was reported by the mean values. The difference directly attributed to VAP in the average hospital stay (patient-day) is an increase by an average of 14 days (63.6%), and the costs of hospital treatment reported an average increase of €1918.00 (74.7%). Several studies conducted in the NICU and PICU [5,7,12,22,23], as well as those involving adult patients [24,25,26], reported that most of the costs associated with VAP were due to an increase in the hospital stay. In a 2-year study conducted with patients in the PICU, VAP was independently associated with increased costs by monitoring the impact of other variables in association with costs, including age, underlying disease, days of mechanical ventilation and severity of disease [21].

The results of the conducted economic analysis are in accordance with already published studies that have highlighted that VAP increases hospital stay and costs [12,27,28]. However, our data show a much larger increase in the hospital stay (63.6%) and costs (74.7%) attributed to VAP than previously reported. In addition, we observe almost identical average hospital costs per day for neonates with and without VAP (€125.78 vs. €118.08), which suggests that the increased number of patient-days is the main driver of increasing the costs. 

The observed longer LOS (patient-days) and days on mechanical ventilation are confirmed by other authors for neonatal and pediatric VAP patients [27,29], with a tendency to increase mortality [7,30,31], which in our case does not reach statistical significance between the two groups of patients [32].

VAP is proven to be the leading NI in the NICU, not only in our prospective study, but in the retrospective period from 2012 to 2016 as well [33]. This infection proves to be the most problematic for the studied setting and the reasons for that might be complex. This is one of the leading NICUs in the country in which neonates in very severe conditions from 8 (out of 28) districts have been managed [33,34].

Prevention of nosocomial infections, including VAP, is based on strategies to reduce the susceptibility of newborns to infections by limiting risk factors and strengthening the body’s defenses. One of the most important preventive measures when it comes to VAP is early extubation, the use of a closed endotracheal suctioning system and switching to non-invasive ventilation methods, such as NCPAP. Several studies have shown a reduction in the VAP rate after guidelines’ implementation into a bundle [35,36]. The power of the bundle is that it brings together several evidence-based practices that individually improve care, but when applied together, may result in an even greater improvement in the desired outcome [37].

However, our research has several limitations, including the design, which is a case-control without adjustment. Thus, we were not able to adjust for patients who were outborn in our analysis, because the information about external use for our study setting is not available. The data are not publicly available; therefore, we were not able to access them. It is logical that this specific group impacts the number of patient-days recorded and the potential costs and adds to the overall burden upon the healthcare system, especially the National Health Insurance Fund, but the initial expenses were calculated for different settings which are not in the focus of our study. Another limitation is that not matching the cases might not eliminate confounding; although, speaking about epidemiological case-control studies, there are records that the result was almost the same, and identical results were found irrespective of whether matching or not matching was applied [38]. In addition, this study was conducted in one hospital, but it is located in the second largest Bulgarian city and moreover, the NICU of this hospital is the only available option for the South-Central region population, which represents approximately 20.0% of the overall Bulgarian population [14]. We could consider that the “St. George” NICU’s resources, staff and patient numbers are similar to the NICUs located in other five Bulgarian regions. Ideally, sampling across other NICUs in Bulgaria would have supported the case for the generalizability of the findings. Summarizing the results of an economic analysis is often a challenge, as actual costs vary across institutions based on different staff costs and different models of supply and use of pharmaceutical products. Yet, the use of costs rather than clinical pathways is preferable, as costs are considered a more reliable assessment of the financial burden, and more accurately describe institutional comparisons. We believe that our study somewhat underestimates the real costs of VAP, as its narrow perspective analyzes direct costs and does not include costs for medication. In addition, indirect costs such as the economic burden on the family due to loss of income, family break-up and costs of pain and/or disability are not included because of methodological issues and lack of information. No attempt has been made to measure the impact of functional deficits in patients with VAP. 

## 5. Conclusions

This study is exclusively targeted at neonatology patients and can be used for a comparative analysis of data from other similar wards. This is the first attempt to estimate the economic impact of VAP in a NICU in a Bulgarian setting. VAP remains a serious and unresolved problem in pediatric and neonatal intensive care units. VAP is particularly associated with prematurity, low birth weight and prolonged mechanical ventilation. Our analysis confirms the results of other studies that the increased length of hospital stay due to VAP results in an increase in hospital costs.

## Figures and Tables

**Table 1 healthcare-10-00980-t001:** Comparison of patient characteristics in the study by groups.

Variables	Cases with VAP (*n* = 33)	Cases without VAP (*n* = 74)	*p*-Value
Demographic characteristics			
Males, *n* (%)	20 (60.6)	40 (54.1)	0.632 ^1^
Age at admission (days), median 25th, 75th percentile (min-max)	1; 1, 1 (1–5)	1; 1, 1 (1–5)	0.813 ^2^
Weight (g) median; 25th, 75th percentile (min–max)	1310; 965, 2400 (570–3700)	1690; 1208, 2730 (530–4660)	0.045 ^2^
Gestational age (weeks) mean ± SD (min–max)	31.1 ± 4.8 (25–41)	33.1 ± 4.3 (23–40)	0.035 ^3^
Clinical data			
APGAR score 1st min. mean ± SD (min–max)	4.74 ± 2.61 (0–9)	5.01 ± 2.56 (1–9)	0.620 ^3^
APGAR score 5th min. mean ± SD (min–max)	7.77 ± 2.56 (0–10)	8.01 ± 1.92 (0–10)	0.590 ^3^
Prematurity based on gestational age *n*, (%)			
32–37 g.w.	5 (18.5)	23 (44.2)	0.024 ^1^
28–32 g.w.	9 (33.3)	15 (28.8)	0.679 ^1^
>28 g.w.	13 (48.1)	14 (27.0)	0.061 ^1^
Time of admission			
0 to 6 h after birth	29 (87.9)	63 (85.1)	0.700 ^1^
6 to 12 h after birth	2 (6.1)	5 (6.8)	0.893 ^1^
12 to 24 h after birth	1 (3.0)	2 (2.7)	0.931 ^1^
>24 h after birth	1 (3.0)	4 (5.4)	0.587 ^1^
Age at VAP diagnosis median; 25th, 75th percentile (min–max)	8; 7,10 (2–32)	-	-
LOS (days) median; 25th, 75th percentile (min–max)	32; 19, 46 (10–100)	18; 11, 27 (2–87)	<0.001 ^2^
LOS (days) before VAP diagnosis median; 25th, 75th percentile (min-max)	8; 6.5, 10.5 (3–25)	-	-
Mechanical ventilation (days) median; 25th, 75th percentile (min–max)	12; 8.25, 24.50 (4–49)	4; 3, 7 (2–23)	<0.001 ^2^
Mechanical ventilation before VAP diagnosis median; 25th, 75th percentile (min–max)	8; 6.5, 10.5 (2–26)	-	-
Lethality *n* (%)	6 (18.2)	16 (21.6)	0.688 ^1^
Costs			
Overall hospital costs (€) median; 25th, 75th percentile (min–max)	3675.77; 2498.87, 5146.35 (1114.06–14,230.96)	2327.78; 1434.10, 3226.83 (243.93–10,818.69)	<0.001 ^2^
Costs for antibiotic therapy (€) median; 25th, 75th percentile (min–max)	432.79; 282.48, 994.23 (147.73–2994.17)	351.61; 212.42, 587.75 (52.60–1675.55)	0.024 ^2^

^1^ z-test for comparison of two percentages; ^2^ Mann–Whitney test; ^3^
*t*-test for comparison of 2 arithmetic means of independent samples.

**Table 2 healthcare-10-00980-t002:** Microorganisms isolated in patients with VAP.

Microorganisms	Number of Isolated Microorganisms in Patients with VAP *n* (%)
Gram-positive microorganisms	
*Enterococcus faecalis*	4 (6.1)
*Enterococcus faecium*	1 (1.5)
*Coagulase (-) Staphylococcus*	1 (1.5)
Overall Gram-positive microorganisms	6 (9.1)
Gram-negative microorganisms	
*Klebsiella pneumoniae ESBL+*	18 (27.3)
*Acinetobacter baumannii*	9 (13.7)
*Pseudomonas aeruginosa*	8 (12.1)
*Escherichia coli*	8 (12.1)
*Klebsiella oxytoca ESBL+*	4 (6.1)
*Strenotrophomonas maltophilia*	4 (6.1)
*Enterobacter cloacae ESBL+*	2 (3.0)
*Chryseobacterium indologenes*	2 (3.0)
*Enterobacter aglomerans*	1 (1.5)
*Acinetobacter lwoffii*	1 (1.5)
*Chryseobacterium gleum*	1 (1.5)
Overall Gram-negative microorganisms	60 (90.9)
Overall isolates	66 (100)

**Table 3 healthcare-10-00980-t003:** Financial impact attributed to ventilator-associated pneumonia: additional costs for VAP.

	Cases with VAP (*n* = 33)	Cases without VAP (*n* = 74)		
Overall LOS (days)	1178	1611		
Overall days of antibiotic therapy (days)	1029	1364		
Overall hospital costs (€)	148,133	190,180		
Overall antibiotic costs (€)	2,4061	33,586		
Directly attributed to VAP difference				
			Number	%
Average length of stay (days)	36	22	14	63.6
Average duration of antibiotic treatment (days)	31	18	13	72.2
Average hospital costs (€)	4489	2569	1918	74.7
Average costs for antibiotics (€)	729	454	273	60.1
Average hospital costs per day (€)	125	118	7	5.9
Average costs for antibiotics per day (€)	23.5	24.5	−1	−4.0

## Data Availability

Not applicable.

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
