# Peer review of "Cost Analysis for Patients with Ventilator-Associated Pneumonia in the Neonatal Intensive Care Unit"

_healthcare, 2022, doi:10.3390/healthcare10060980_

Round 1

Reviewer 1 Report

Major comments

  1. I wonder whether the authors considered modeling the costs using descriptive variables found to be different between groups as adjustments in regression analysis.

Minor comments

  1. Abstract (line 28). Please use p<0.001 for p-values below this value.

  1. Table 1 (line 173). Please use p<0.001 for p-values below this value.

  1. Table 2 (line 185). Consider ordering the categories by the number of isolated microorganisms (within each Gram-type) for a more intuitive interpretation of the results.

  1. Results (line 196). Please use p<0.001 for p-values below this value.

Author Response

We would like to thank the reviewer for the insightful and helpful comments on our manuscript. We have been able to incorporate changes to reflect most of the suggestions provided. Following are the responses to the comments given:

Comment 1: I wonder whether the authors considered modelling the costs using descriptive variables found to be different between groups as adjustments in regression analysis.

Response: Thank you for this particular comment. Yes, we did consider modeling the costs and this is in the focus of another manuscript that is going under peer-review process. The National Health Insurance Fund payer’s perspective was analyzed.

Comment 2: Abstract (line 28). Please use p<0.001 for p-values below this value.

Response: We do agree with the comment. We have revised the whole manuscript and changed p=0.000 to p<0.001.

Comment 3:  Table 1 (line 173). Please use p<0.001 for p-values below this value.

Response: We do agree with the comment. We have revised the whole manuscript and changed p=0.000 to p<0.001.

Comment 4: Table 2 (line 185). Consider ordering the categories by the number of isolated microorganisms (within each Gram-type) for a more intuitive interpretation of the results.

Response: Thank you for this insightful recommendation. We have revised the table ordering the microorganisms by the number of isolated pathogens.

Comment 5:  Results (line 196). Please use p<0.001 for p-values below this value.

Response: We do agree with the comment. We have revised the whole manuscript and changed p=0.000 to p<0.001.

Reviewer 2 Report

Cost analysis for patients with ventilator-associated pneumonia
in the neonatal intensive care unit is an interesting article submitted by Ralitsa Raycheva , Vanya Rangelova, and Ani Kevorkyan. The study comes from a large regional NICU taking care of 20% of population of the nation where the study is done. The article is very well written. However the following clarifications would help.

The causes of increased cost could be due to the following added reasons which is not clarified.

  1. Was there increased use of total parenteral nutrition in such patients?
  2. Was there an increased incidence of shock in this population? Did that contribute to increased inotrope use leading to prolonged hospital stay
  3. It will be curious to know about the suction practice in the NICU as more frequent suctioning may be associated with reduced VAP.

Author Response

We would like to thank the reviewer for the insightful and helpful comments on our manuscript. Following are the responses to the comments given:

Comment 1: Was there increased use of total parenteral nutrition in such patients?

Response:  Thank you for this comment. There was increased use of total parenteral nutrition in patients with VAP, mainly due to the significantly longer hospital stay. As we don’t have information for all the non-VAP patients on the use of parenteral nutrition we haven’t included this in the manuscript.

Comment 2: Was there an increased incidence of shock in this population? Did that contribute to increased inotrope use leading to prolonged hospital stay?

Response: In the group of patients with VAP, there was not an increased incidence of shock after diagnosing VAP, as these were neonates born with combined neonatology pathology and inotropes were used shortly after birth to overcome the shock. After VAP diagnosis the use of inotropes was not significantly higher compared to the control group of non-VAP patients.

Comment 3: It will be curious to know about the suction practice in the NICU as more frequent suctioning may be associated with reduced VAP.

Response: Thank you for this thoughtful comment. Endotracheal intubation was performed by observing the standard precautions (sterile gowns, masks, laryngoscope blades and tubes) to ensure maintaining sterility of equipment until use. Endotracheal suctioning was performed every 8 hours in case of a need for microbiological material for examination. Closed systems for endotracheal suctioning were used in the NICU. We have added this clarification in the section Materials and Methods as well.